# Adaptation and validation of the Polish version of the Beliefs about Medicines Questionnaire among cardiovascular patients and medical students

Michał Seweryn Karbownik[1]*, Beata Jankowska-Polańska[2], Robert Horne[3], Karol Maksymilian Górski[1], Edward Kowalczyk[1], Janusz Szemraj[4]

1 Department of Pharmacology and Toxicology, Medical University of Lodz, Łódź, Poland, 2 Department of Clinical Nursing, Wroclaw Medical University, Wrocław, Poland, 3 Centre for Behavioural Medicine, The School of Pharmacy, University College London, London, United Kingdom, 4 Department of Medical Biochemistry, Medical University of Lodz, Łódź, Poland

* michal.karbownik@umed.lodz.pl

## Abstract

### Background

The Beliefs about Medicines Questionnaire (BMQ) is the leading tool intended to assess the cognitive representation of medication, however, the validated Polish version of the questionnaire is lacking.

### Aims

To adapt the original BMQ tool to the Polish language (BMQ-PL) and to validate it.

### Materials and methods

The BMQ was adapted to Polish according to widely accepted guidelines. A total of 311 cardiovascular in- and outpatients as well as medical students taking chronic medication were surveyed to assess data-to-model fit and internal consistency of the measure. The criterion-related validity was determined with the use of Polish version of the Adherence to Refills and Medications Scale. Confirmatory and exploratory factor analyses were used, as well as general linear modeling.

### Results

The BMQ-PL exhibited the same factorial structure as the original questionnaire and all the items loaded on their expected factors. Internal consistency of the questionnaire was satisfactory in the group of cardiovascular patients (Cronbach's alpha ranging from 0.64 to 0.82 and McDonald's omega from 0.90 to 0.91). There were significant correlations in the predicted directions between all BMQ-PL subscales and the measure of drug adherence in cardiovascular outpatients, but not in inpatients. Medical students may conceptualize the

**Data Availability Statement:** The raw data underlying the process of BMQ validation are

available in Mendeley Data repository (http://dx.doi.org/10.17632/rvxcb9p2md.1).

**Funding:** This work was supported by the Medical University of Lodz, Poland (https://umed.pl/) with the grant received by JS (503/6-086-01/503-61-001-19-00). The funder had no role in study design, data collection and analysis, decision to publish, or preparation of the manuscript.

**Competing interests:** The authors have declared that no competing interests exist.

beliefs about medicines in a different way; as a result, a modified version of the BMQ-PL-General, suitable for medically-educated people, was proposed.

## Conclusion

The BMQ-PL exhibits satisfactory proof of validity to be used among cardiovascular patients.

## Introduction

The effectiveness of drug therapy is not only a function of the pharmacological properties of the drug and the physiology of the patient, but also adherence to a drug regimen [1]. It is estimated that almost a half of chronically-ill patients do not follow the prescribed dosing regimen [2–4], making non-adherence one of the most important factors contributing to treatment failure [5–8], and many other unfavorable social and economic effects [2, 9]. Understanding and overcoming patient non-adherence is considered a major goal in healthcare [8] and the subject of improving adherence has attracted much attention [10].

There are many factors shaping adherence to a treatment protocol. One is based on the beliefs of the patients about their medications [10–12]. Many studies have found a negative perception of medications on the part of the patient to be a barrier to adherence [11]. For example, Horne and Weinman [13] report that having fewer concerns about taking prescribed medications and stronger perceptions of their necessity are positively associated with the self-reported medication adherence in groups of chronically-ill patients. These results were confirmed by a meta-analysis of data from more than 25,000 patients [14] and the use of a more objective measure of medication adherence calculated by pharmacy dispensing records [15]. Also, studies based on qualitative methods have confirmed the importance of patient concerns about medications as a key barrier to adherence [16–17]. Moreover, acknowledging patient concerns about medications and positive reinforcement of the need for medication have the potential to improve adherence and quality of life among patients [18].

The conceptualization and operationalization of patient beliefs about medications is challenging. However, a measure intended to assess the cognitive representation of medication, namely the Beliefs about Medicines Questionnaire (BMQ), was developed by Horne et al. [19]. The measure is well theoretically and scientifically grounded, supported by multiple qualitative and quantitative studies. It is intended to self-assess commonly-held beliefs about medicines, being adequate for patients suffering from a range of chronic illnesses. The questionnaire is an 18-item measure requiring assessment of each item in a 5-point Likert scale (Fig 1). The BMQ has become the leading tool for measuring beliefs about medicines. The paper reporting the development of the questionnaire has already been cited 1,779 times (according to *Google Scholar*–February 13, 2020) and the questionnaire was adapted to multiple languages [20]. The BMQ has already been used in Poland, however, the process of its adaptation to and validation in Polish language has not been reported [21], and no official version of the questionnaire exists. Indeed, it is of ultimate importance to ensure the validity of the used measures in order to confirm integrity of any reported research outcomes [22].

Therefore, the aim of this study was to adapt the original BMQ tool to the Polish language (BMQ-PL) and to validate it.

## Materials and methods

The adaptation of the BMQ to Polish was guided by Tsang et al. [24].

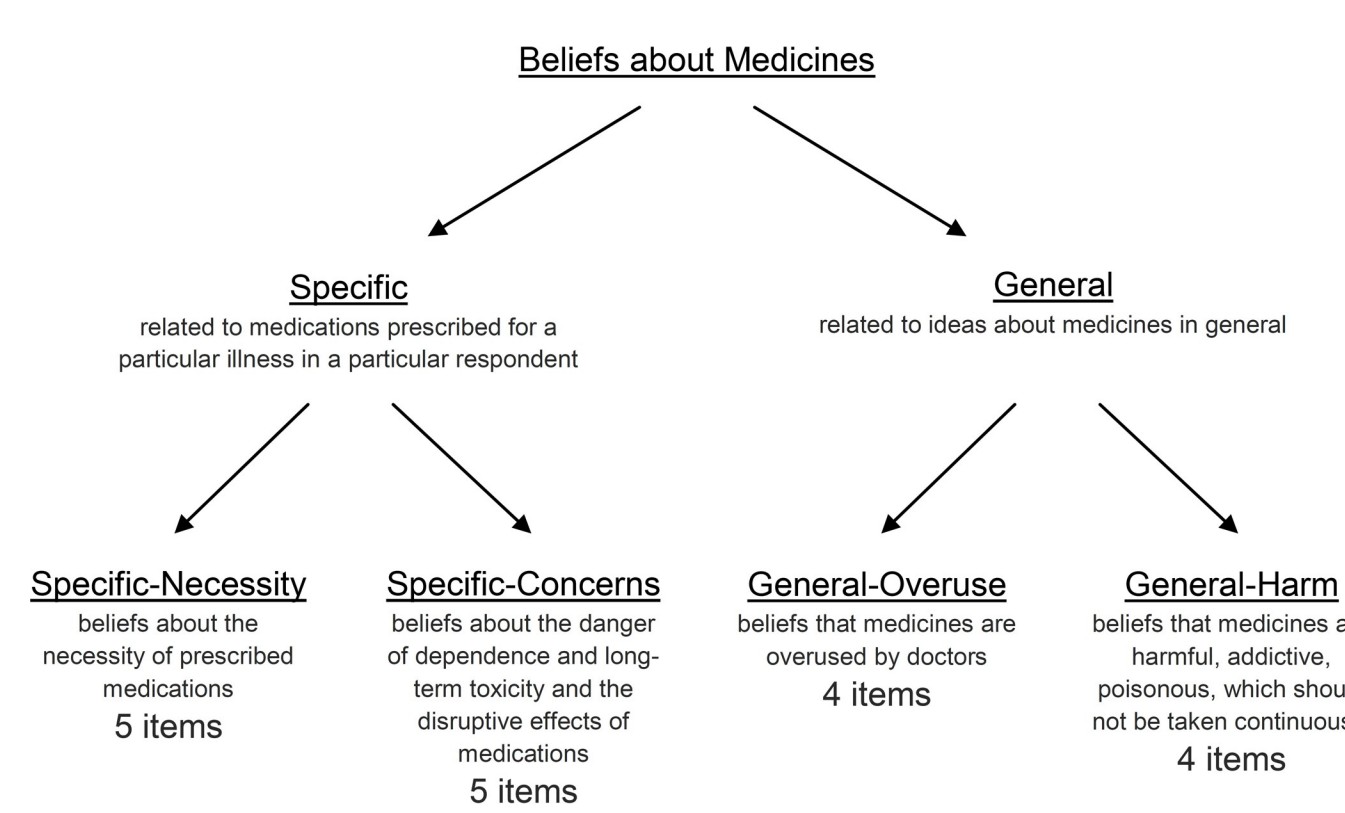

**Fig 1. Composition of the Beliefs about Medicines Questionnaire.** More recent versions have added an item to the *Specific-Concerns* subscale and a 4-item *Benefit* subscale of the BMQ-General creating a total of 23-item questionnaire; however, no full validation of such tool has been performed [23].

### Ethical considerations

The study was carried out in agreement with the latest version of the Declaration of Helsinki. It was approved by the Bioethics Committee of the Medical University of Lodz (number RNN/319/17/KE received on October 17, 2017). All the questionnaires were gathered from May 2018 to March 2019. In all cases, written or electronic informed consent to participate was given by the respondent.

### Questionnaire forward translation

Permission to translate and validate the BMQ was obtained from prof. Robert Horne, the leading developer of the original English version of the questionnaire [19]. The original BMQ was independently translated to Polish by 2 native speakers of Polish who are proficient in English: a medical doctor and a pharmacist, both with academic and clinical experience. To support accuracy and unambiguity of the final translated version, several fully validated different language versions of the BMQ were used for reference during the translation process, particularly the French [25] and Scandinavian [26].

### Review of the translated version

A panel of Polish native experts was formed to discuss the forward translations of the BMQ. The panel included 2 BMQ translators (see above) and, additionally, a psychologist, an academic English language teacher and a medical student. The panel thoroughly considered and discussed the semantics of the introduction to the questionnaire, each of the 18 statements, and rating scale labels. Afterwards, a single translated version was chosen. This version was

reviewed by a specialist in Polish linguistics and further modified according to her suggestions.

Six people were then interviewed. All were non-medically educated, elderly general or cardiovascular patients with no apparent cognitive impairment; 4 women and 2 men took part, mean age 67.5 ± 11.9 years; their level of education ranged from primary to higher. The interview was performed in a semi-structured way according to general guidelines [27] and a protocol for questionnaire pretesting [28]. The aim of the interview was to assess the degree of understanding of the introduction, each of the 18 statements, and the rating scale labels of the translated questionnaire. The feedback received was used to construct the final Polish version of the Beliefs about Medicines Questionnaire (BMQ-PL, © prof. Robert Horne) and to complement further quantitative analysis. Written informed consent was received from all participants before taking part.

## Questionnaire back-translation

A British native speaker who is a professional English language editor at the Medical University of Lodz, who lives in Poland and speaks Polish well, was asked to do a back-translation of the BMQ-PL (Polish to English). The translator did not participate in the expert panel and was blinded to the content of the original BMQ. After completing the back-translation, the translator was shown the original BMQ and was asked to compare it with a back-translated version of BMQ-PL in order to report the potential discrepancies of the meaning.

## Understanding of the BMQ-PL

The BMQ-PL was administered in electronic form (Google Forms, Google, Mountain View, CA, USA) to a group of 14 people (8 non-medically educated general patients and 6 medical students: 6 were women, mean age 38.6 ± 19.0 years), who were asked in a structured way how much each statement in the questionnaire is understandable (unambiguous and raising no doubts) and readable (easy and enjoyable to read). A 5-point Likert scale with anchors of "not at all" (1) to "very much" (5) was used to gather the responses. The respondents were also asked to provide their own understanding of each statement in the questionnaire. Informed consent was received in electronic form from all participants.

## Readability of the BMQ-PL

To further assess the readability of the BMQ-PL, the introduction to the questionnaire together with all the statements were subjected to analysis using *Jasnopis* (https://jasnopis.pl/): a tool to assess the difficulty of a Polish language text on a 7 degree scale [29].

## Further validation of the questionnaire

Further analyses were performed to validate the BMQ-PL questionnaire: to assess and compare the data-to-model fit among Polish cardiovascular patients and medical students and to define the internal consistency of the BMQ-PL. The criterion-related validity of the measure was also determined by examining the relationship with self-reported drug adherence using the Polish version of the Adherence to Refills and Medications Scale (ARMS) [30]. The surveys also included sex, age, nationality, place of residence, education and number of medications used.

For the purpose of further validation of the BMQ-PL, the following groups of patients were recruited: 1) cardiovascular inpatients, 2) cardiovascular outpatients and 3) medical students who report chronic medicines use.

1) Cardiovascular inpatients were recruited on admission to the clinical wards of Department of Angiology, Hypertension and Diabetology, as well as Department and Clinic of Internal and Occupational Diseases and Hypertension, Wroclaw Medical University, Poland. A medical doctor assessed whether a patient meets inclusion or exclusion criteria (see below) and a qualified nurse (the same across the whole study, who was not a part of the healthcare team) introduced the assumptions of the study. Patients were informed that the hospital healthcare staff would stay blinded to the results of the questionnaires. After receiving written informed consent, the surveys were distributed. The patients were asked to complete their surveys themselves, but the nurse was available if help was needed with any questions regarding the study. Sociodemographic and clinical data was obtained from the medical records following data anonymization by attribute suppression.

Inclusion criteria were:

- diagnosis of arterial hypertension in accordance with the current guidelines of European Society of Hypertension

- treatment with at least 1 antihypertensive drug for 6 months or longer

- age of 18 years or more.

Exclusion criteria were:

- exacerbation of concurrent severe chronic diseases

- any mental disorder confirmed with medical record

- cognitive impairment (Mini-Mental State Examination, MMSE above 23 points)

- inability to communicate in Polish

- a sight defect that does not allow reading the questionnaire

- refusal of written informed consent.

2) Cardiovascular outpatients were recruited from universities of the third age (U3As): 2 located in Łódź and 2 in Sosnowiec. The use of such sources of participants allowed drug adherence and beliefs about medicines to be examined in non-medical settings, which could have minimized observer-expectancy bias. During a regular meeting of the members of the U3As a 10-minute introductory speech about beliefs about medicines and drug adherence was delivered, and the assumptions of the study together with inclusion and exclusion criteria (see below) were presented and discussed. The initial speech was planned to underline scientific interest in drug adherence and beliefs about medicines, without affecting the way in which the participants respond to the survey questions. After receiving written informed consent, the surveys were distributed. The respondents completed their surveys themselves, however, a researcher was available while the surveys were being completed to respond to any questions asked by study participants.

Inclusion criteria were:

- self-reported cardiovascular disease (particularly arterial hypertension, ischemic heart disease, heart failure or arrhythmia)

- self-reported taking of at least 1 cardiovascular drug for a year or longer

Exclusion criteria were:

- apparent cognitive impairment (reported by study participant or researcher)

- inability to communicate in Polish

- a sight defect that does not allow reading the questionnaire

- refusal of written informed consent.

3) Medical students were recruited through the social media of Department of Pharmacology and Toxicology, Medical University of Lodz, Poland. Only medical and dental students of the third year or above were invited. The students were requested to fill out the electronic survey. In the introduction to the survey, the assumptions of the study together with inclusion and exclusion criteria (see below) were presented. Information was provided about the possibility to contact the main investigator by e-mail regarding the study. After confirming that all the inclusion criteria were met, that no exclusion criteria were met, and that electronic informed consent had been received, the survey was displayed.

Inclusion criteria were:

- being a medical or dental student of the third year or above in the Medical University of Lodz

- self-reported chronic use (defined as at least 90 days a year) of at least 1 medication for a year or longer (preparations used for contraceptive purpose were not defined as "medications" in this study)–the students who did not report chronic medication use were allowed to participate in the study and to complete the BMQ-General part of the questionnaire, but only the data of those meeting the criterion of chronic medication use was included into the analyses

- age of 18 years or more.

Exclusion criteria were:

- exacerbation of concurrent severe chronic diseases

- inability to communicate in Polish

- refusal of electronic informed consent.

## Further adaptation of the BMQ-General for medical students

As the factorial structure representation in the group of medical students was unsatisfactory, further steps were taken to adapt the questionnaire to this group. Firstly, a separate group of medical students were approached in the same way as described above and, following receiving informed consent, they were applied electronic survey with all the statements included in the BMQ-PL. The students were asked to attribute each *Specific* statement to the category of *Necessity*, *Concerns* or *None of the indicated* and each *General* statement to the category of *Overuse*, *Harm* or *None of the indicated*. Following this, 6 medical students or young medical doctors were interviewed in a semi-structured way regarding the understanding of each of the BMQ item according to general guidelines [27] and a protocol for questionnaire pretesting [28]. Thirdly, the newly proposed item composition of BMQ-PL-General-Med scale was applied to the data of medical students taking drugs chronically. Fourthly, the BMQ-PL-General-Med item composition was externally validated in the group of medical students who did not report chronic medication use (see above).

## Data analysis

Missing data accounted for 0.77% of the analyzed database and was handled with pairwise deletion. Descriptive statistics included means and standard deviations (SD) or absolute and

relative frequencies, if not stated otherwise. Confirmatory factor analysis (CFA) was used to determine the extent to which the obtained data fits into the assumed structure of the BMQ and multiple fit indices were estimated. Multi-group confirmatory factor analysis (MGCFA) was performed to test measurement invariance between the groups of patients [31]: configural, metric, scalar and residual variance invariance were successively tested by increasing levels of group equality constraints imposed on factor loadings, item intercepts and residual variances. The analyses were performed separately for each pair of the tested groups: cardiovascular inpatients vs. outpatients, cardiovascular inpatients vs. medical students, and cardiovascular outpatients vs. medical students. A $\chi^2$ difference test was used to compare the hierarchical models. Exploratory factor analysis (EFA) was used to further analyze the individual performance of observed variables in predicting latent variables. Cronbach's alpha and McDonald's omega were used to estimate scale internal consistency. Although the Likert scale data used in the questionnaires should be perceived as ordinal variables, the associations between the examined constructs were assessed with Pearson correlation coefficient (r) and were further explored with general linear modeling (GLM), i.e. parametric tests. This was to allow for multivariate modeling. However, corresponding equivalent non-parametric tests were performed where possible. Testing multiple hypotheses was not a subject for significance level correction because of the explorative and preliminary nature of the research. *P*-values were presented with 2 significant figures and rounded to a maximum of 4 decimal places. *P*-values below 0.05 were considered statistically significant. The analysis was performed using STATISTICA 13.1 Software (StatSoft, Tulsa, OK, USA). The raw data underlying the process of BMQ validation was deposited in Mendeley Data repository (https://doi.org/10.17632/rvxcb9p2md.1).

## Results

### Understanding and readability of the BMQ-PL

The Polish version of BMQ was proposed as a result of the original BMQ forward translation and its thorough review. Back-translation of the BMQ-PL identified minor semantic differences from the original questionnaire. The results of the semi-structured interviews of 6 elderly patients indicated that the BMQ-PL is understood as assumed by the investigators, and no such modifications in questionnaire content were needed. However, the interview identified a few misconceptions of the BMQ-PL content, which helped explain the phenomena revealed by the further quantitative analysis (see S1 File).

The next electronic structured interview of 14 respondents indicated that the mean score for understanding and readability of each item was more than 4 (on the 5-point Likert scale). Analysis of the BMQ-PL with the *Jasnopis* tool graded the questionnaire difficulty as 3 out of 7, which is interpreted by the tool as "easy text, understandable to an average Pole". A sentence-by-sentence comparison of the original BMQ, BMQ-PL, and its back translation as well as the estimates of understanding and readability, is presented in Table 1.

### Study participants in the further validation process

Out of 165 cardiovascular inpatients invited to the study, 36 did not meet the inclusion criteria (31 because of an unacceptably low MMSE score), and a further 16 declined to participate in the study (12 because of unwillingness to fill the survey). In total, 113 surveys were distributed; however, 6 people resigned from participation in the study at this step, and 5 surveys were excluded due to being incorrectly filled. Hence, 102 surveys of cardiovascular inpatients were included to the study (drop-out rate of 9.7%).

Out of 157 attendees of the U3As invited to the study, 32 (20.4%) declared not using any cardiovascular medication for a year or longer, and further 17 declined to participate in the

**Table 1. Comparison of the original, Polish, and back-translated version of the BMQ.** Understanding and readability were estimated using a 5-point Likert scale with anchors of "not at all" (1) to "very much" (5).

| No | Original | Polish | Back-translated | Difference | Understanding* (mean ± SD) | Readability** (mean ± SD) |
|---|---|---|---|---|---|---|
| Questionnaire title | | | | | | |
| | Beliefs about medicines questionnaire | *Kwestionariusz Przekonań na temat Leków* | Questionnaire on opinion on drugs | No significant difference | N/A | N/A |
| Specific scale–introduction | | | | | | |
| | Your views about medicines prescribed for you. | *Twoje poglądy o stosowanych przez Ciebie lekach* | Your opinion on drugs that you were administered. | No significant difference | N/A | N/A |
| | We would like to ask you about your personal views about medicines prescribed for you. | *Chcielibyśmy zapytać Cię o Twoje osobiste poglądy na temat stosowanych przez Ciebie leków.* | We would like you to give us your personal opinion on drugs that you are administered. | No significant difference | N/A | N/A |
| | These are statements other people have made about their medicines. | *Poniżej znajdują się stwierdzenia innych ludzi na temat leków, które stosują.* | Below, there are other people's opinions on drugs that they are administered. | No significant difference | N/A | N/A |
| | Please indicate the extent to which you agree or disagree with them by ticking the appropriate box. | *Proszę, wskaż, w jakim stopniu zgadzasz się z tymi stwierdzeniami stawiając krzyżyk w odpowiednim kwadracie.* | Please, state how much you agree on these opinions by putting a tick in the appropriate box. | No significant difference | N/A | N/A |
| | There are no right or wrong answers. We are interested in your personal views. | *Nie ma dobrych ani złych odpowiedzi. Jesteśmy zainteresowani Twoimi własnymi poglądami.* | There are neither good nor bad opinions. We are interested in your personal opinions. | No significant difference | N/A | N/A |
| Rating scale labels | | | | | | |
| | strongly agree, agree, uncertain, disagree, strongly disagree | *zdecydowanie zgadzam się, zgadzam się, nie jestem pewien / pewna, nie zgadzam się, zdecydowanie nie zgadzam się* | I completely agree, I agree, I am not sure, I disagree, I completely disagree | No significant difference | N/A | N/A |
| Specific statements | | | | | | |
| S1 | My health, at present, depends on my medicines. | *Moje zdrowie zależy obecnie od leków, które przyjmuję.* | My health currently depends on the medications I take. | No significant difference | 4.64 ± 0.84 | 4.79 ± 0.58 |
| S2 | Having to take these medicines worries me. | *Martwi mnie, że muszę przyjmować te leki.* | I am worried that I have to take these medications. | No significant difference | 4.71 ± 0.83 | 4.93 ± 0.27 |
| S3 | My life would be impossible without my medicines. | *Moje życie byłoby niemożliwe bez leków.* | My life would be impossible without medication. | Lack of a second "my" makes the back-translation less personal. Polish language does not use possessive adjectives so much. | 4.57 ± 0.94 | 4.79 ± 0.58 |
| S4 | Without my medicines I would be very ill. | *Bez moich leków byłbym bardzo chory / byłabym bardzo chora.* | Without my medication I would be very sick. | Back-translation is slightly less formal. In Polish language does not clearly distinguish "ill" and "sick". | 4.71 ± 0.73 | 4.93 ± 0.27 |
| S5 | I sometimes worry about long-term effects of my medicines. | *Czasami martwię się, jakie mogą być długofalowe skutki działania moich leków.* | Sometimes I worry about the long-term effects of my medications. | No significant difference | 4.93 ± 0.27 | 4.79 ± 0.80 |
| S6 | My medicines are a mystery to me. | *Moje leki są dla mnie zagadką.* | My medication is a mystery to me. | No significant difference | 4.43 ± 0.85 | 4.71 ± 0.47 |
| S7 | My health in the future will depend on my medicines. | *W przyszłości stan mojego zdrowia będzie zależał od leków.* | In the future, my health will depend on medicines. | No significant difference | 4.86 ± 0.36 | 4.79 ± 0.43 |
| S8 | My medicines disrupt my life. | *Leki, które przyjmuję, zakłócają moje życie.* | The medication I'm taking interferes with my life. | No significant difference | 4.50 ± 1.09 | 4.36 ± 1.08 |

*(Continued)*

**Table 1.** (Continued)

| No | Original | Polish | Back-translated | Difference | Understanding* (mean ± SD) | Readability** (mean ± SD) |
|---|---|---|---|---|---|---|
| S9 | I sometimes worry about becoming too dependent on my medicines. | *Czasami martwię się, że mogę za bardzo uzależnić się od leków.* | Sometimes I worry that I may be too dependent on my medications. | Back-translation refers to the present whereas an original sentence to the future. | 4.64 ± 0.50 | 4.79 ± 0.43 |
| S10 | My medicines protect me from becoming worse. | *Leki chronią mnie przed pogorszeniem stanu zdrowia.* | Medicine prevents my health from deteriorating. | Back-translation is less specific about the medicines. | 4.57 ± 0.85 | 4.93 ± 0.27 |
| General scale–introduction | | | | | | |
| | Your views about medicines in general | *Twoje poglądy o lekach w ogóle* | Your general opinion on drugs | Polish equivalent of "medicines" used in the questionnaire has no double meaning as "drugs". | N/A | N/A |
| | We would like to ask you about your personal views about medicines in general. | *Chcielibyśmy zapytać Cię o Twoje osobiste poglądy na temat leków.* | We would like you to give us your personal opinion on drugs. | Polish equivalent of "medicines" used in the questionnaire has no double meaning as "drugs". | N/A | N/A |
| | These are statements other people have made about medicines in general. | *Poniżej znajdują się stwierdzenia innych ludzi na temat leków.* | Below, there are other people's general opinions on drugs. | Polish equivalent of "medicines" used in the questionnaire has no double meaning as "drugs". | N/A | N/A |
| | Please indicate the extent to which you agree or disagree with them by ticking the appropriate box. | *Proszę, wskaż, w jakim stopniu zgadzasz się z tymi stwierdzeniami stawiając krzyżyk w odpowiednim kwadracie.* | Please, state how much you agree on these opinions by putting a tick in the appropriate box. | No significant difference | N/A | N/A |
| | There are no right or wrong answers. We are interested in your personal views. | *Nie ma dobrych ani złych odpowiedzi. Jesteśmy zainteresowani Twoimi własnymi poglądami.* | There are neither good nor bad opinions. We are interested in your personal opinions. | No significant difference | N/A | N/A |
| Rating scale labels | | | | | | |
| | strongly agree, agree, uncertain, disagree, strongly disagree | *zdecydowanie zgadzam się, zgadzam się, nie jestem pewien / pewna, nie zgadzam się, zdecydowanie nie zgadzam się* | I completely agree, I agree, I am not sure, I disagree, I completely disagree | No significant difference | N/A | N/A |
| General—statements | | | | | | |
| G1 | Doctors use too many medicines. | *Lekarze przepisują zbyt wiele leków.* | Doctors prescribe too many drugs. | The verb "prescribe" instead of "use" clarifies what the doctors do with the medicines. In fact, such a meaning is suggested in the paper describing the development of the tool [19] and all the Scandinavian versions have also replaced this verb [26]. | 4.50 ± 0.65 | 4.79 ± 0.43 |
| G2 | People who take medicines should stop their treatment for a while every now and again. | *Osoby przyjmujące leki powinny od czasu do czasu robić przerwy w ich stosowaniu.* | People taking medicine should stop doing so from time to time. | No significant difference | 4.29 ±1.07 | 4.71 ± 0.47 |
| G3 | Most medicines are addictive. | *Większość leków uzależnia.* | Most medicines are addictive. | No difference | 4.79 ± 0.43 | 4.93 ± 0.27 |
| G4 | Natural remedies are safer than medicines. | *Naturalne środki lecznicze są bezpieczniejsze niż leki.* | Natural remedies are safer than medicine. | No significant difference | 4.93 ± 0.27 | 5.00 ± 0.00 |
| G5 | Medicines do more harm than good. | *Leki powodują więcej szkody niż pożytku.* | Medications cause more harm than good. | No significant difference | 4.79 ± 0.58 | 4.86 ± 0.36 |
| G6 | All medicines are poisons. | *Wszystkie leki to trucizny.* | All medicines are poisons. | No difference | 4.64 ± 0.84 | 4.64 ± 0.84 |

(*Continued*)

**Table 1.** (Continued)

| No | Original | Polish | Back-translated | Difference | Understanding* (mean ± SD) | Readability** (mean ± SD) |
|---|---|---|---|---|---|---|
| G7 | Doctors place too much trust on medicines. | *Lekarze za bardzo polegają na lekach.* | Doctors rely too heavily on medications. | No significant difference | 4.71 ± 0.61 | 4.93 ± 0.27 |
| G8 | If doctors had more time with patients they would prescribe fewer medicines. | *Gdyby lekarze poświęcali więcej czasu pacjentom, przepisywaliby mniej leków.* | If doctors spent more time with patients, they would prescribe fewer medications. | The verb "spend" instead of "have time" implies that it is a doctor's decision to have more time with patients, but not e.g. a problem with health care system functioning. A French versions has also a similar verb [25]. | 4.79 ± 0.43 | 4.86 ± 0.36 |

* being unambiguous and raising no doubts.

** being easy and enjoyable to read.

SD–standard deviation.

N/A–not applicable.

S1-10 –subsequent *Specific* items, G1-8 –subsequent *General* items.

study. Of the 108 surveys distributed, 6 were found to be incorrectly completed and were excluded. Hence, 102 surveys of cardiovascular outpatients were included to the study (dropout rate of 5.6%).

Out of 403 medical and dental students of the third year or above, who gave electronic informed consent, 40 (9.9%) reported not using medications at all, and 256 (63.5%) reported using medications from time to time, but no more than 90 days a year. A total of 107 out of 403 students (26.6%) reported using medications chronically (at least 90 days a year) and their responses were included to the analyses. Among these 107 students, 42 (39.3%) reported using anti-allergic medications, 30 (28.0%)–analgesics, 26 (24.3%)–hormonal medications, 16 (15.0%)–antibiotics, 14 (13.1%)–psychotropic medications, 13 (12.1%)–pulmonologic medications, 11 (10.3%)–cardiologic medications and 4 (3.7%)–diabetes medications.

The number of respondents in each group was satisfactory based on the recommendations regarding sample size. A minimum of hundred is advised to be sufficient to allow convergence of fit indices [32]. This number is also consistent with the computed sample size: as the root mean square error of approximation (RMSEA) in the population was 0.08, the RMSEA of the null hypothesis was 0.05, significance level was 0.05 and 129 degrees of freedom were used according to the model identification criteria, a sample size of 112 was needed in the study to reach a power of 0.8.

The basic sociodemographic characteristics of the study participants are presented in Table 2.

## Comparisons of the models

Four models were compared in each of the groups of patients to assess the degree of data-to-model fit: 1-factor model (all items loading 1 dimension), 2-factor model (distinguishing *Specific* and *General* dimensions), 3-factor model (distinguishing *Specific-Necessity*, *Specific-Concerns* and *General* dimensions), and 4-factor model (the original model consisting of *Specific-Necessity*, *Specific-Concerns*, *General-Overuse* and *General-Harm* dimensions).

As expected, the 2-factor model exhibited a more acceptable model fit than the 1-factor model ($\Delta\chi^2(1)$ = 98.5, $p$<0.0001 for inpatients, $\Delta\chi^2(1)$ = 60.8, $p$<0.0001 for outpatients, and $\Delta\chi^2(1)$ = 100.7, $p$<0.0001 for medical students), and the 3-factor model exhibited a more acceptable model fit than the 2-factor model ($\Delta\chi^2(2)$ = 99.8, $p$<0.0001 for inpatients, $\Delta\chi^2(2)$ =

**Table 2. Basic sociodemographic characteristics of the study participants.**

| Variable | Number (frequency) or mean (standard deviation) | | |
|---|---|---|---|
| | Cardiovascular patients | | Medical students (n = 107) |
| | Inpatients (n = 102) | Outpatients (n = 102) | |
| Sex | | | |
| Male | 45 (44.1%) | 15 (14.7%) | 31 (29.0%) |
| Female | 57 (55.9%) | 87 (85.3%) | 76 (71.0%) |
| Age* | | | |
| [years] | 63.2 (14.3) range: 20–98 | 73.1 (7.4) range: 55–89 | 23.0 (1.4) range: 22–29 |
| Nationality | | | |
| Polish | 102 (100%) | 102 (100%) | 107 (100%) |
| other | 0 (0%) | 0 (0%) | 0 (0%) |
| Place of residence | | | |
| < 5,000 inhabitants | 21 (20.6%) | 3 (2.9%) | 23 (21.5%) |
| 5,000–50,000 inhabitants | 21 (20.6%) | 15 (14.7%) | 29 (27.1%) |
| 50,000–500,000 inhabitants | 15 (14.7%) | 19 (18.6%) | 28 (26.2%) |
| > 500,000 inhabitants | 45 (44.1%) | 65 (63.7%) | 27 (25.2%) |
| Education** | | | |
| Primary | 5 (5.0%) | 10 (10.0%) | 0 (0%) |
| Secondary | 69 (68.3%) | 53 (53.0%) | 105 (98.1%) |
| Higher | 27 (26.7%) | 37 (37.0%) | 2 (1.9%) |
| How many medications do you use?*** | | | |
| 1 | 4 (4.6%) | 32 (31.4%) | 42 (39.3%) |
| 2 | 10 (11.5%) | 30 (29.4%) | 38 (35.5%) |
| 3 | 19 (21.8%) | 16 (15.7%) | 18 (16.8%) |
| 4 | 21 (24.1%) | 18 (17.6%) | 9 (8.4%)**** |
| 5 or more | 33 (37.9%) | 6 (5.9%) | |

* In the group of outpatients n = 96.

** In the group of inpatients n = 101, and outpatients n = 100.

*** In the group of inpatients n = 87. In this group the number of medications used refers to the time before hospitalization.

**** "4 medications or more"–the question in the group of medical students did not include the option "5 or more".

The proportion of males to females varied significantly across the groups of participants ($\chi^2(2) = 21.3$, $p < 0.0001$).

73.6, $p < 0.0001$ for outpatients, and $\Delta\chi^2(2) = 65.5$, $p < 0.0001$ for medical students). Interestingly, the original 4-factor model fit was slightly more favorable than that of the 3-factor model only in the group of medical students ($\Delta\chi^2(3) = 14.7$, $p = 0.0021$), but not in either inpatients ($\Delta\chi^2(3) = 4.7$, $p = 0.20$) or outpatients ($\Delta\chi^2(3) = 5.9$, $p = 0.12$). Moreover, Parsimony-corrected Comparative Fit Index as well as Akaike's and Bayesian Information Criteria (Table 3) favored the simpler, 3-factor model, over the original 4-factor model for both inpatients and outpatients. The differences between discussed indices were so slight, however, that they could potentially disappear with follow-up study. Moreover, the majority of the cross-cultural BMQ validation literature supports the use of the 4-factor model; this is also the case for German and Czech languages, which are spoken in countries bordering Poland and culturally similar to Poland. As no solid evidence can be found against using the two-dimensional *General* beliefs about medicines represented by Polish people, the original 4-factor model of the BMQ-PL was preserved.

The fit indices of 4-factor model for the group of inpatients were satisfactory. The group of outpatients presented slightly worse model fit, not meeting conventional fit criteria. The model fit for the group of medical students was far from conventional fit criteria. The comparisons of the models and their fit indices are presented in the Table 3. The results of the 4-factor model confirmatory factor analysis are presented in Fig 2.

## Measurement invariance between the groups of patients

To further explore equivalency of the 4-factor BMQ model between the groups of patients, MGCFA analysis was performed. The detailed results of the MGCFA are presented in S2 File. The results are briefly discussed below.

1) Configural invariance was tested to assess whether the different groups of participants conceptualize the constructs in the same way. The comparison of in- and outpatients gives a model with the fit indices slightly worse than conventional cut-off criteria. Hence, the characteristics of both groups of cardiovascular patients appear somewhat different. On the other hand, medical students may employ a substantially different mode of construct conceptualization than the two groups of cardiovascular patients. For the purpose of this study, however, further steps of measurement invariance testing were performed in all pairs of comparisons.

2) Metric invariance was tested to assess whether the different groups of participants respond to the items in the same way, attributing them the same meaning. The criteria for metric invariance was met only for the comparison of in- and outpatients, and for outpatients vs. medical students. The full metric invariance test of cardiovascular inpatients vs. medical students failed, but the criteria for partial metric invariance between these groups were met for the *Specific-Concerns* and *General-Harm* subscales, both considering negative attributes of medicines.

3) Scalar invariance was tested to assess whether the different groups of participants have the same basal level of observed variables (intercepts). Testing full scalar invariance resulted in the rejection of the hypothesis of scalar invariance across all the groups. However, the criteria for partial scalar invariance were met for *Specific-Necessity* and *General-Overuse* subscales for the comparison of in- and outpatients: both these subscales consider use and overuse of medicines. The results suggest that the basal level of responses displayed by the groups of respondents significantly depends on their social and clinical state.

4) Residual variance invariance was tested to assess whether the different groups of participants present the same level of unexplained variance in responses to the items. This type of invariance was established only for the pair of groups, which presented at least partial scalar invariance, i.e. cardiovascular in- and outpatients. The test implies that the residual variance is not invariant between these groups, apart from the *Specific-Concerns* subscale. This suggests the confounders play important differential role in the way how the participants respond to the items.

## Internal consistency and other psychometric properties

Internal consistency estimates, measured as Cronbach's alpha, were acceptable ($\geq$0.7) [34] for the *Specific-Necessity* subscale in all groups, as well as for *Specific-Concerns* in inpatients and *General-Overuse* in outpatients. The other Cronbach's alpha values were questionable ($\geq$0.6) with the *General-Harm* subscale in medical students being particularly low. On the other hand, McDonald's omega estimates of internal consistency were good in all groups (Table 4).

In order to further analyze the extent to which the individual item participates in latent variable determination, item discrimination (measured as a corrected item-total correlation with assumed scales) was calculated. Great majority of the item discrimination values were at least

**Table 3. Confirmatory factor analysis model fit parameters for 1-, 2-, 3-, and 4-factor models.**

| Model | Group of patients | | |
|---|---|---|---|
| | Inpatients | Outpatients | Medical students |
| **1-factor model** | | | |
| $\chi^2$ (df) | 384.7 (135) | 332.2 (135) | 398.2 (135) |
| *P*-value | <0.0001 | <0.0001 | <0.0001 |
| $\chi^2$/df | 2.85 | 2.46 | 2.95 |
| RMSEA (90% CI) | 0.164 (0.149–0.180) | 0.142 (0.126–0.158) | 0.163 (0.148–0.179) |
| TLI | 0.427 | 0.464 | 0.319 |
| CFI/PCFI | 0.495 / 0.358 | 0.529 / 0.369 | 0.400 / 0.288 |
| AIC / BIC | 4.52 / 5.46 | 4.26 / 5.23 | 4.52 / 5.44 |
| SRMR | 0.135 | 0.140 | 0.157 |
| **2-factor model** | | | |
| $\chi^2$ (df) | 286.2 (134) | 271.4 (134) | 297.5 (134) |
| *P*-value | <0.0001 | <0.0001 | 2.22 |
| $\chi^2$/df | 2.14 | 2.03 | <0.0001 |
| RMSEA (90% CI) | 0.114 (0.098–0.131) | 0.114 (0.096–0.131) | 0.122 (0.106–0.138) |
| TLI | 0.647 | 0.623 | 0.572 |
| CFI/PCFI | 0.692 / 0.489 | 0.672 / 0.460 | 0.627 / 0.435 |
| AIC / BIC | 3.57 / 4.53 | 3.64 / 4.64 | 3.57 / 4.52 |
| SRMR | 0.143 | 0.147 | 0.129 |
| **3-factor model** | | | |
| $\chi^2$ (df) | 186.4 (132) | 197.8 (132) | 232.0 (132) |
| *P*-value | 0.0013 | 0.0002 | <0.0001 |
| $\chi^2$/df | 1.41 | 1.50 | 1.76 |
| RMSEA (90% CI) | 0.054 (0.027–0.076) | 0.070 (0.047–0.090) | 0.091 (0.073–0.108) |
| TLI | 0.870 | 0.815 | 0.733 |
| CFI/PCFI | 0.890 / 0.615 | 0.843 / 0.564 | 0.772 / 0.524 |
| AIC / BIC | 2.62 / 3.63 | 2.90 / 3.96 | 2.98 / 3.98 |
| SRMR | 0.079 | 0.091 | 0.096 |
| **4-factor model** | | | |
| $\chi^2$ (df) | 181.7 (129) | 191.9 (129) | 217.3 (129) |
| *P*-value | 0.0016 | 0.0003 | <0.0001 |
| $\chi^2$/df | 1.41 | 1.49 | 1.68 |
| RMSEA (90% CI) | 0.055 (0.027–0.077) | 0.067 (0.044–0.088) | 0.082 (0.063–0.100) |
| TLI | 0.871 | 0.819 | 0.759 |
| CFI/PCFI | 0.894 / 0.607 | 0.850 / 0.560 | 0.799 / 0.533 |
| AIC / BIC | 2.63 / 3.72 | 2.90 / 4.04 | 2.90 / 3.97 |
| SRMR | 0.079 | 0.090 | 0.093 |

$\chi^2$ –chi-square statistics.

df–degrees of freedom.

CI–confidence intervals.

RMSEA–Root Mean Square Error of Approximation. The strict cut-off point estimate close to 0.06 with a lower 90% CI limit close to 0 and the upper limit less than 0.08 is currently considered "a good fit". In the past, however, the recommendations were less strict: a point estimate below 0.08 was considered "a good fit", whereas between 0.08 to 0.10 "a mediocre fit".

TLI–Tucker-Lewis index or Non-Normed Fit Index. Values at least 0.95 are preferred, but values as low as 0.80 were also acceptable.

CFI–Comparative Fit Index. Values at least 0.95 are presently recognized as indicative of "a good fit", but the limit of 0.90 was proposed in the past.

PCFI–Parsimony-corrected Comparative Fit Index. While no threshold levels have been recommended for parsimony-corrected indices, it is suggested that the values of at least 0.50 or, even better, 0.60 should be obtained.

AIC–Akaike's Information Criterion.

BIC–Bayesian Information Criterion. Both AIC and BIC are used for assessment relative to other models. The smaller the values, the better and more parsimonious the model fit.

SRMR–standardized root mean square residual. Values less than 0.05 represent "a good fit", however, values as high as 0.08 are deemed "acceptable".

The above recommendations for the model fit indices were reported in Hooper et al. [33].

0.2, indicating adequate loading of the latent variables. The mean values for the response to each item were close to 3, the middle value in the BMQ 5-point Likert scale. The dispersion of the responses were adequate (Table 5).

Exploratory factor analysis was used in further analysis of the construct representation by the items. The analysis was performed in each group of patients separately for *Specific* and *General* scales. The number of factors in the analyses was rigidly fixed as 2 for *Specific* and 2 for *General* scales in order to reflect the assumed structure of the constructs. In fact, such a number of extracted factors was consistent with the scree-plot and the eigenvalue-more-than-one criterion. The extracted factors could represent slightly more than 50% of data variability. The analysis indicated that the items representing the *Specific* facets of beliefs have adequate primary factor loadings with almost no substantial cross-loadings. On the other hand, the factor loadings of majority of items representing *General* facets of beliefs were diverse, not clearly indicating whether the particular item maps *Overuse* or *Harm* subscale (Table 6).

Four latent constructs comprising beliefs about medicines were significantly inter-correlated. Particularly, *General-Overuse* with *General-Harm*, and *Specific-Concerns* with both *General* subscales. *Specific-Necessity* was found to negatively correlate with both *General* subscales only in the group of inpatients, however, its association with *General-Harm* was uncertain due to non-significant result of a non-parametric test (Table 7).

## Criterion-related validity of the questionnaire

The criterion-related validity of the BMQ-PL questionnaire was assessed with regard to the self-reported measure of medication adherence (Polish version of the ARMS scale). It was assumed that adherence is positively associated with *Specific-Necessity*, and negatively with *Specific-Concerns* and both *General* subscales. *Necessity-minus-Concerns*, calculated as a difference between the scores of the subscales [14], was assumed to be the most indicative of medication adherence. In fact, the hypotheses worked well for the group of outpatients, but in the group of inpatients, the assumed association approached significance level only in the *Specific-Concerns* subscale. The adherence was not monitored in the group of medical students due to the diverse patterns of medication use in this group (Table 8 and Fig 3).

The number of medications used was positively correlated with the following BMQ-PL subscales (after combining patient groups): *Specific-Necessity* ($r = 0.33$, $p < 0.0001$), *Specific-Concerns* ($r = 0.32$, $p < 0.0001$), and *General-Harm* ($r = 0.19$, $p = 0.0012$), but not with *General-Overuse* ($r = 0.01$, $p = 0.88$). Results of equivalent non-parametric tests for these associations were presented in S1 Table and their similarity to parametric ones documented in S1 Fig.

## Further adaptation of the BMQ-General for medically-educated respondents

Following a series of qualitative and quantitative analyses, a modified version of the BMQ-PL-General was proposed, which was suitable for medically-educated respondents (BMQ-PL-General-Med). The scale consisted of all the original items except G6 (*All medicines are poisons*), grouped into a 3-item *Overuse* subscale (items G1, G7, G8) and a 4-item *Harm* subscale (items G2, G3, G4, G5). The scale exhibited good data-to-model fit, even in external validation, and satisfactory internal consistency. The details on the analyses performed and their results are reported in S3 File.

## Discussion

In order to accurately measure and report patient beliefs about medicines, a rigorously corroborated tool is needed [22]. The present study is the first to describe the Polish version of the

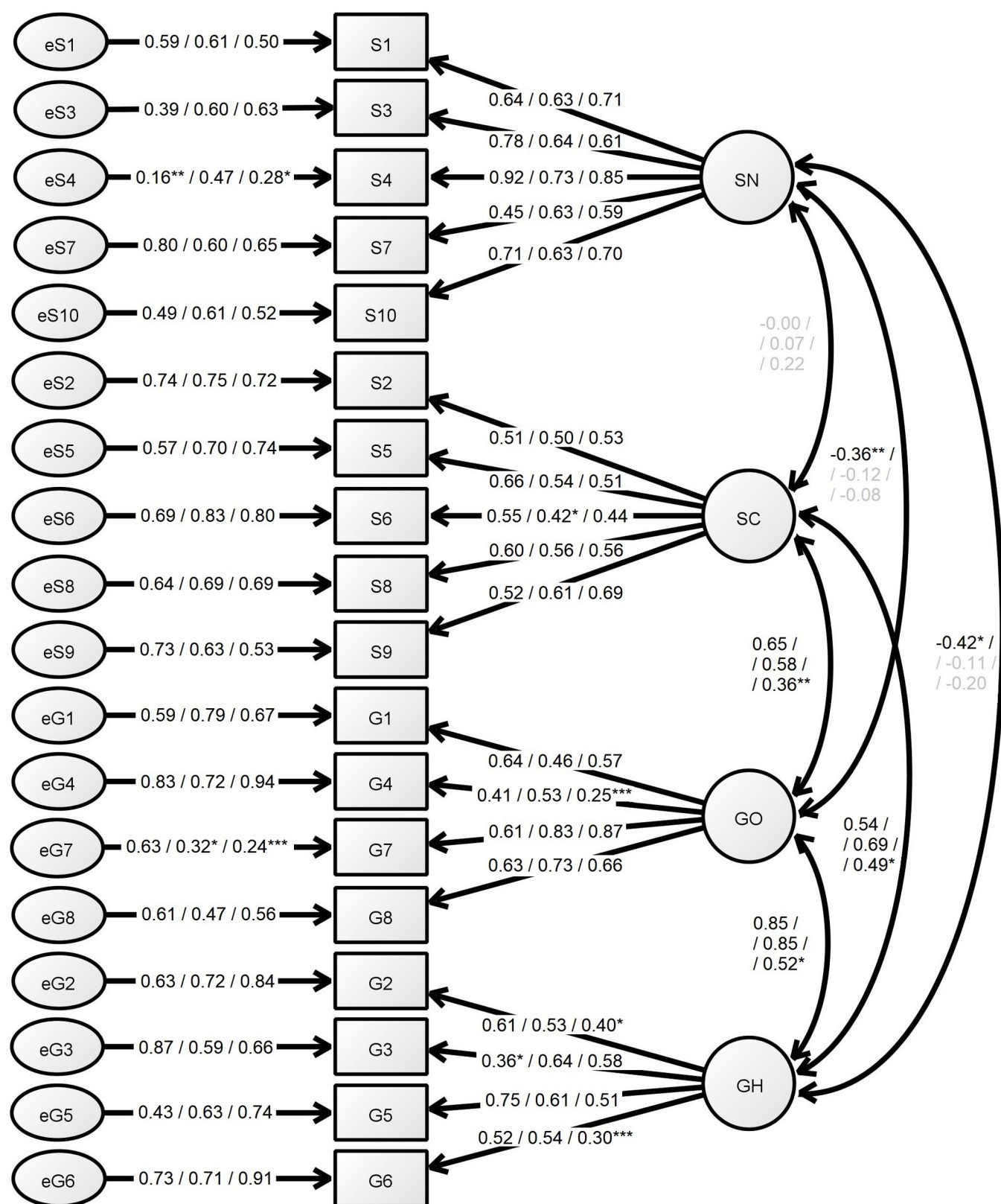

**Fig 2. Confirmatory factor analysis of the 4-factor model of the BMQ-PL.** The numerical values represent lambda coefficients of the indicators for the groups of Inpatients / Outpatients / Students. The $p$-values for the significance of the parameters are <0.0001, if not stated otherwise: $^*p$ = 0.0001–0.001,

**p = 0.001–0.01, ***p = 0.01–0.05, lambda coefficients presented in grey are non-significant (p≥0.05). SN–*Specific-Necessity* subscale, SC–*Specific-Concerns* subscale, GO–*General-Overuse* subscale, GH–*General-Harm* subscale, S1-10 –subsequent *Specific* items, G1-8 –subsequent *General* items, e–latent unobserved error related to the measurement.

Beliefs about Medicines Questionnaire (BMQ-PL) and provide satisfactory data for its validity. The original questionnaire was adapted to Polish with an in-depth lexical analysis. Its structure was recognized in the group of cardiovascular in- and outpatients as well as in medical students taking chronic medication.

The process of English-to-Polish translation of the original BMQ involved a few substantial modifications in the item wording. This was presented and discussed in detail in Table 1. The decision to reword some of the items stemmed from the detailed analysis of an article describing the development of the original questionnaire [19], this was supported by other adaptations of the questionnaire [25, 26] and finally endorsed by the main Author of the original tool. The BMQ-PL was found to transmit a clear message in the target language and, ultimately, was acceptably well understood and readable by Polish patients. The presence of a few notions regarding the way the BMQ-PL content was understood do not undermine its appropriateness, but rather shed light on explaining the phenomena reported with quantitative analysis.

The selection of the groups of patients for the validation analysis was not accidental. Cardiovascular diseases are the number 1 cause of death globally [35], and cardiovascular patients represent considerable proportion of the elderly society. According to the recent report of the Central Statistical Office in Poland [36], 48.8% of people aged 60–69 take cardiovascular medications, whereas among people aged 70 and more, the proportion reaches as much as 72.7%. It was decided not to include the patients with other diseases to the study, as similar questionnaire structure representation and proof of validity was expected. In fact, in the original BMQ development and validation report, the factor structure exhibited an acceptable degree of stability between groups of asthmatic, diabetic, renal, cardiac, psychiatric and general medical patients, confirming its versatility [19]. The process of validation of the French version of the BMQ was performed in the group of patients suffering from such a diverse diseases as diabetes and AIDS, and the authors concluded that the construct manifests in the same way in each disease group [25]. In contrast, it was decided to compare cardiovascular patients suffering from their disease in the opposite circumstances: hospitalized patients in the day of admission, whose health may be endangered and not stable (inpatients), with the socially-active participants of U3As surveyed in non-medical settings (outpatients). In- and outpatients may not substantially differ in terms of their knowledge about the disease [37], but may differ in terms of disease advance, control and comorbidities [38], consequently representing very different groups of patients. Additionally, the group of medical students taking medications chronically (at least 90 days a year) was included into the analysis. Despite obvious differences, this group

**Table 4. Internal consistency of the BMQ subscales.**

| Internal consistency measures | | Group of patients | | |
|---|---|---|---|---|
| | | **Inpatients** | **Outpatients** | **Medical students** |
| Cronbach's alpha | *Specific-Necessity* | 0.82 | 0.79 | 0.82 |
| | *Specific-Concerns* | 0.70 | 0.65 | 0.66 |
| | *General-Overuse* | 0.66 | 0.70 | 0.66 |
| | *General-Harm* | 0.64 | 0.65 | 0.42 |
| McDonald's omega | | 0.91 | 0.91 | 0.90 |

**Table 5. Basic psychometric characteristics of the BMQ-PL items.**

| Subscale / / Item | Group of patients | | | | | |
|---|---|---|---|---|---|---|
| | Inpatients | | Outpatients | | Medical students | |
| | Item discrimination | Mean (SD) | Item discrimination | Mean (SD) | Item discrimination | Mean (SD) |
| *Specific-Necessity* | | 19.9 (3.0) | | 19.1 (3.8) | | 16.2 (5.2) |
| S1 | 0.62 | 4.1 (0.8) | 0.56 | 3.9 (1.1) | 0.64 | 3.6 (1.3) |
| S3 | 0.65 | 3.8 (0.9) | 0.57 | 3.6 (1.1) | 0.52 | 2.4 (1.4) |
| S4 | 0.80 | 4.0 (0.8) | 0.63 | 3.7 (1.0) | 0.75 | 2.8 (1.5) |
| S7 | 0.39 | 3.8 (0.8) | 0.55 | 3.8 (1.0) | 0.57 | 3.4 (1.3) |
| S10 | 0.67 | 4.2 (0.7) | 0.53 | 4.2 (0.9) | 0.62 | 4.0 (1.2) |
| *Specific-Concerns* | | 15.7 (3.6) | | 14.2 (3.9) | | 10.9 (3.7) |
| S2 | 0.40 | 3.6 (1.1) | 0.39 | 3.1 (1.3) | 0.42 | 3.2 (1.4) |
| S5 | 0.54 | 3.6 (0.9) | 0.40 | 3.4 (1.0) | 0.48 | 3.2 (1.4) |
| S6 | 0.39 | 2.8 (1.1) | 0.32 | 2.7 (1.2) | 0.35 | 1.4 (0.7) |
| S8 | 0.50 | 2.9 (1.1) | 0.42 | 2.3 (1.2) | 0.43 | 1.5 (0.8) |
| S9 | 0.45 | 2.9 (1.1) | 0.49 | 2.7 (1.3) | 0.49 | 1.7 (1.1) |
| *Necessity-minus-Concerns* | | 4.2 (4.6) | | 5.1 (5.2) | | 5.3 (5.8) |
| *General-Overuse* | | 13.0 (2.8) | | 12.9 (3.6) | | 12.2 (3.5) |
| G1 | 0.45 | 3.2 (1.1) | 0.34 | 2.9 (1.3) | 0.45 | 3.3 (1.2) |
| G4 | 0.31 | 3.1 (0.9) | 0.40 | 3.1 (1.2) | 0.20 | 2.7 (1.2) |
| G7 | 0.51 | 3.1 (1.0) | 0.61 | 3.2 (1.2) | 0.65 | 2.7 (1.3) |
| G8 | 0.48 | 3.5 (1.0) | 0.61 | 3.7 (1.3) | 0.52 | 3.4 (1.3) |
| *General-Harm* | | 11.1 (2.6) | | 10.1 (3.1) | | 7.3 (2.7) |
| G2 | 0.49 | 2.9 (1.0) | 0.37 | 2.9 (1.2) | 0.24 | 2.0 (1.2) |
| G3 | 0.35 | 3.1 (1.0) | 0.42 | 3.0 (1.1) | 0.32 | 1.9 (1.1) |
| G5 | 0.55 | 2.6 (0.8) | 0.51 | 2.0 (0.9) | 0.36 | 1.5 (0.8) |
| G6 | 0.32 | 2.5 (0.9) | 0.43 | 2.2 (1.2) | 0.09 | 1.9 (1.3) |

SD–standard deviation.

S1-10 –subsequent *Specific* items, G1-8 –subsequent *General* items.

Rating scale labels: strongly agree– 5, agree– 4, uncertain– 3, disagree– 2, strongly disagree– 1.

may be similar to the cardiovascular outpatients due to their shared academic background. The assessment of beliefs about medicines in medical students and healthcare professionals has already been performed [39–41], but only little attention was paid to ensure the validity of the tool in such a group of respondents [42] and to measure invariance between people with and without medical education. The current report complements these shortcomings.

The analysis presented in this study confirmed that the construct of beliefs about medicines has a 4-factor structure, identical with the original version. However, an alternative 3-factor structure, combining all the *General* beliefs, is possible as well. This is reflected also by the substantial association between the *General-Overuse* and *General-Harm* subscales in all the tested groups of patients (Pearson's r ranging from 0.45 to 0.61). Such association was replicated in many other samples tested with the BMQ: the original one (r = 0.40) [19], the French (r = 0.55 and r = 0.87) [25], the German (r = 0.49) [43] and the Greek language versions (r = 0.36) [44] suggesting that both *General* concepts might be merged and termed simply *General-Negative*. Such a proposal of 3-factor BMQ representation was reported for Spanish psychiatric outpatients [42]. However, there is little evidence against the 4-factor structure in the Polish cardiovascular patients, and therefore it is highly advisable to use it in further Polish research.

**Table 6. Results of exploratory factor analysis performed in each scale separately.** a) *Specific* scale, b) *General* scale. The factor differentiation was achieved by raw varimax factor rotation. The expected loadings are presented in bold on a dark-grey background, with substantial cross-loadings on a light-grey background.

| | | Group of patients | | | | | |
|---|---|---|---|---|---|---|---|
| | | Inpatients | | Outpatients | | Medical students | |

### a) *Specific* scale

| Subscale | Item | Factor loadings of expected subscale | | | | | |
|---|---|---|---|---|---|---|---|
| | | Specific-Necessity | Specific-Concerns | Specific-Necessity | Specific-Concerns | Specific-Necessity | Specific-Concerns |
| Specific-Necessity | S1 | **0.77** | -0.04 | **0.74** | -0.14 | **0.82** | -0.06 |
| | S3 | **0.81** | 0.03 | **0.70** | 0.05 | **0.63** | 0.16 |
| | S4 | **0.89** | -0.01 | **0.77** | -0.02 | **0.84** | 0.16 |
| | S7 | **0.55** | 0.09 | **0.71** | 0.26 | **0.71** | -0.00 |
| | S10 | **0.81** | -0.03 | **0.72** | -0.09 | **0.79** | -0.10 |
| Specific-Concerns | S2 | 0.22 | **0.63** | 0.04 | **0.62** | 0.43 | **0.54** |
| | S5 | 0.03 | **0.75** | 0.14 | **0.63** | -0.02 | **0.68** |
| | S6 | -0.06 | **0.62** | 0.25 | **0.51** | -0.15 | **0.64** |
| | S8 | -0.07 | **0.71** | -0.14 | **0.67** | 0.12 | **0.62** |
| | S9 | -0.11 | **0.66** | -0.07 | **0.76** | 0.01 | **0.78** |
| Eigenvalue | | 3.06 | 2.29 | 2.77 | 2.17 | 3.11 | 2.22 |
| Variance explained | | 30.6% | 22.9% | 27.7% | 21.7% | 31.1% | 22.2% |

### b) *General* scale

| Subscale | Item | Factor loadings | | | | | |
|---|---|---|---|---|---|---|---|
| | | General-Overuse | General-Harm | General-Overuse | General-Harm | General-Overuse | General-Harm |
| General-Overuse | G1 | **0.79** | -0.05 | **-0.00** | 0.82 | **0.62** | 0.32 |
| | G4 | **0.20** | 0.65 | **0.74** | 0.03 | **0.13** | 0.62 |
| | G7 | **0.62** | 0.26 | **0.59** | 0.52 | **0.84** | 0.17 |
| | G8 | **0.71** | 0.12 | **0.47** | 0.59 | **0.85** | -0.06 |
| General-Harm | G2 | 0.39 | **0.64** | 0.21 | **0.71** | -0.02 | **0.71** |
| | G3 | -0.05 | **0.85** | 0.60 | **0.30** | 0.17 | **0.66** |
| | G5 | 0.71 | **0.32** | 0.75 | **0.12** | 0.13 | **0.67** |
| | G6 | 0.60 | **0.07** | 0.52 | **0.25** | 0.50 | **0.06** |
| Eigenvalue | | 2.57 | 1.75 | 2.36 | 1.97 | 2.12 | 1.90 |
| Variance explained | | 32.1% | 21.8% | 29.5% | 24.6% | 26.5% | 23.7% |

S1-10 –subsequent *Specific* items, G1-8 –subsequent *General* items.

*Specific* scale for the group of Inpatients:

KMO = 0.721; the Bartlett's test of sphericity: $\chi^2(45) = 307.2$, $p<0.0001$.

*Specific* scale for the group of Outpatients:

KMO = 0.720; the Bartlett's test of sphericity: $\chi^2(45) = 218.4$, $p<0.0001$.

*Specific* scale for the group of Medical students:

KMO = 0.734; the Bartlett's test of sphericity: $\chi^2(45) = 322.7$, $p<0.0001$.

*General* scale for the group of Inpatients:

KMO = 0.761; the Bartlett's test of sphericity: $\chi^2(28) = 181.6$, $p<0.0001$.

*General* scale for the group of Outpatients:

KMO = 0.812; the Bartlett's test of sphericity: $\chi^2(28) = 193.9$, $p<0.0001$.

*General* scale for the group of Medical students:

KMO = 0.718; the Bartlett's test of sphericity: $\chi^2(28) = 145.7$, $p<0.0001$.

KMO–Kaiser-Meyer-Olkin measure of sampling adequacy.

Some Cronbach's alpha estimates of internal consistency were below the conventional acceptable threshold [34]. However, as the BMQ-PL includes meaningful content and represents reasonable unidimensionality of the subscales [19], low internal consistency may not be a major barrier to its validity [45]. In fact, other BMQ validation studies have also replicated

**Table 7. Association between the BMQ-PL subscales.** The associations are estimated with Pearson's correlation coefficients. Statistically significant results are presented in bold. The extent of cell shading reflects the extent of correlation. Results of equivalent non-parametric tests for all the associations were presented in S1 Table and their similarity to parametric ones documented in S1 Fig.

| | Group of patients | | | | | | | | |
| | Inpatients | | | Outpatients | | | Students | | |
| | SN | SC | GH | SN | SC | GH | SN | SC | GH |
|---|---|---|---|---|---|---|---|---|---|
| SC | 0.01 $p = 0.89$ | | | 0.08 $p = 0.47$ | | | 0.18 $p = 0.071$ | | |
| GH | -0.28* $p = 0.0045$ | **0.39 $p<0.0001$** | | -0.09 $p = 0.40$ | **0.44 $p<0.0001$** | | -0.13 $p = 0.18$ | **0.28 $p = 0.0034$** | |
| GO | **-0.28 $p = 0.0041$** | **0.44 $p<0.0001$** | **0.53 $p<0.0001$** | -0.09 $p = 0.38$ | **0.40 $p<0.0001$** | **0.61 $p<0.0001$** | -0.05 $p = 0.59$ | **0.31 $p = 0.0015$** | **0.45 $p<0.0001$** |

SN–*Specific-Necessity* subscale.

SC–*Specific-Concerns* subscale.

GO–*General-Overuse* subscale.

GH–*General-Harm* subscale.

* equivalent non-parametric Spearman's rho test yielded non-significant result ($p = 0.088$).

such questionable Cronbach's alpha values [19, 25, 46]. Cronbach's alpha is interpreted as the extent of equivalence of different sets of subscale items that give the same measurement outcomes [47]. Consequently, a low value suggests that different items within a subscale are not closely related to each other and may cover different facets of the same construct. It must be also noted that Cronbach's alpha may be negatively biased if certain assumptions, such as equal factor loadings of subscale items or non-correlated errors, are violated. As this may be the case in the present study, McDonald's omega values, which were adequate for the BMQ-PL, appear more robust estimators of internal consistency [34, 48].

Confirmatory factor analysis indicated a satisfactory data-to-model fit for cardiovascular inpatients, strengthening the proof of validity of the BMQ-PL in this sample. The tested factorial structure in the group of cardiovascular outpatients was less represented by the responses to the individual items, which suggests that the cognitive depiction of medications was less coherent in this group. This may result from the diversity of the group of outpatients, whose medical diagnosis, comorbidities, verified list of medications, and social situation were not

**Table 8. Association of the BMQ-PL subscales with self-reported measure of drug adherence.** Drug adherence was assessed with the use of Polish version of the Adherence to Refills and Medications Scale. Statistically significant results are presented in bold. The extent of cell shading reflects the extent of correlation. Results of equivalent non-parametric tests for the univariate associations were presented in S1 Table and their similarity to parametric ones documented in S1 Fig.

| | Group of cardiovascular patients | | | | Difference between in- and outpatients** | |
| | Inpatients | | Outpatients | | | |
| | Raw estimate | Adjusted estimate* | Raw estimate | Adjusted estimate* | Raw *p*-value | Adjusted *p*-value* |
|---|---|---|---|---|---|---|
| *Specific-Necessity* | -0.04 $p = 0.71$ | -0.02 $p = 0.84$ | **0.29 $p = 0.0053$** | 0.22 $p = 0.059$ | **$p = 0.031$** | $p = 0.078$ |
| *Specific-Concerns* | **-0.21*** $p = 0.038$** | **-0.24 $p = 0.036$** | **-0.24 $p = 0.024$** | **-0.24 $p = 0.021$** | $p = 0.85$ | $p = 0.71$ |
| *Necessity-minus-Concerns* | 0.14 $p = 0.18$ | 0.17 $p = 0.13$ | **0.40 $p = 0.0001$** | **0.33 $p = 0.0018$** | $p = 0.069$ | $p = 0.38$ |
| *General-Overuse* | 0.08 $p = 0.45$ | 0.10 $p = 0.39$ | **-0.27 $p = 0.0091$** | **-0.27 $p = 0.0095$** | **$p = 0.024$** | **$p = 0.049$** |
| *General-Harm* | -0.11 $p = 0.27$ | -0.11 $p = 0.35$ | **-0.30 $p = 0.0030$** | **-0.25 $p = 0.015$** | $p = 0.19$ | $p = 0.27$ |

* adjusted for age, sex, education, place of residence and number of drugs used.

** General Linear Model–variables included to the model: group, BMQ subscale and their two-way interaction.

*** equivalent non-parametric Spearman's rho test yielded non-significant result ($p = 0.063$).

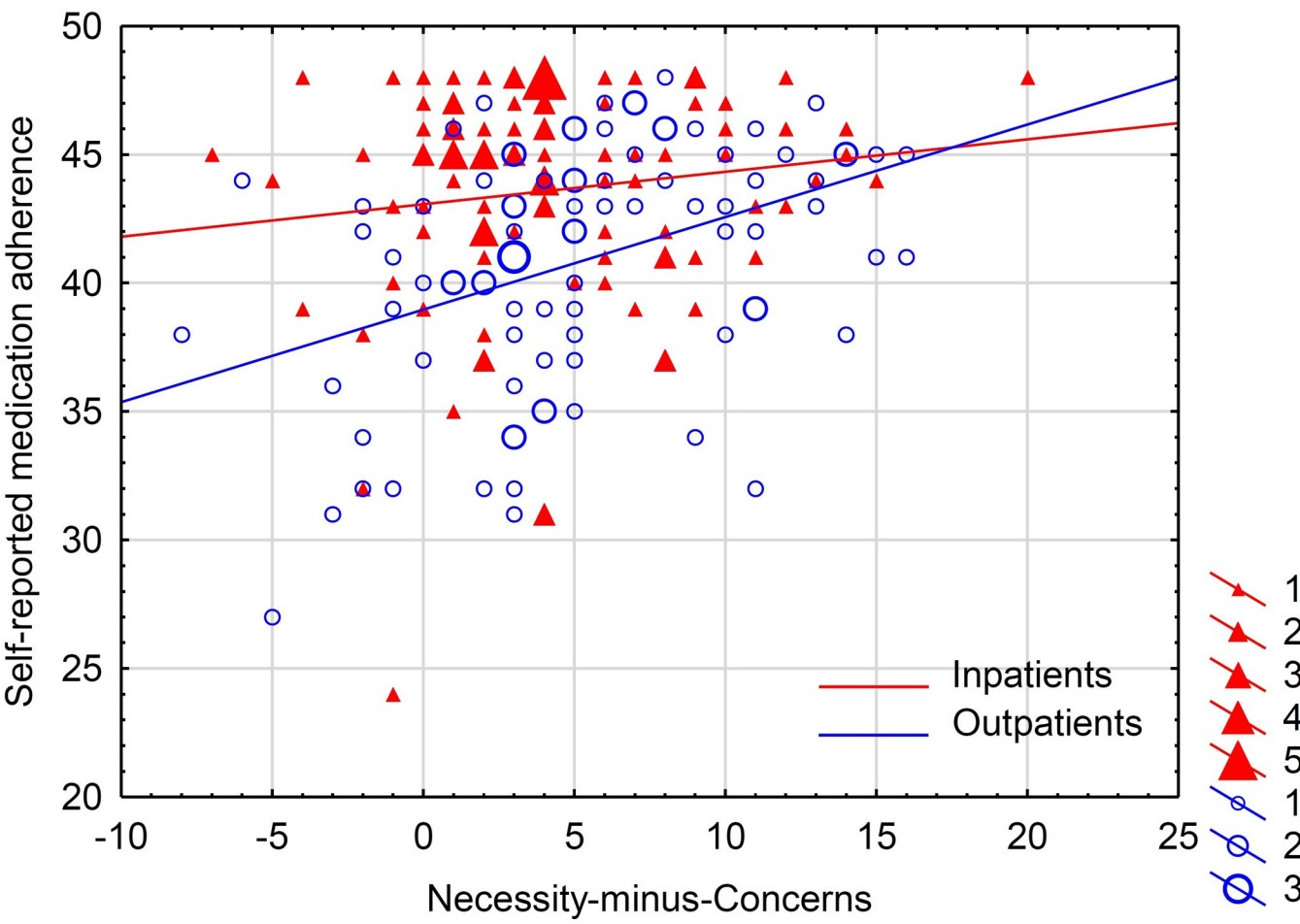

**Fig 3. Association between self-reported medication adherence and *Necessity-minus-Concerns* beliefs about medicines.** Adherence was reported with the use of the Polish version of Adherence to Refills and Medications Scale (ARMS), and *Necessity-minus-Concerns* as a difference between the relevant *Specific* subscales of the Polish version of Beliefs about Medicines Questionnaire (BMQ-PL). The results are reported separately for Inpatients (red colored) and Outpatients (blue colored). The size of the markers represents the number of cases with the given coordinates. Quantitative analysis of the association is reported in Table 8.

confirmed with medical records, and the inclusion and exclusion criteria to participate in the study were largely self-assessed. Nevertheless, this fact does not preclude the possibility of valid use of the BMQ-PL in this group of patients.

The measurement invariance [31] of the construct representation was somewhat confirmed between inpatients and outpatients. Although the groups were found to conceptualize "beliefs about medicines" in slightly different way (configural non-invariance), they attributed the same meaning to individual questionnaire items (metric invariance). However, the background assessment of the observed variables constituting *Specific-Concerns* and *General-Harm* subscales appeared more negative in the group of inpatients than outpatients (scalar non-invariance). This is not surprising, as the sense of deteriorating health and the personal burden of hospitalization or polypharmacy [49]–as compared to relative well-being of socially active attendees of the U3A [50]–may intensify medication-related concerns and beliefs about their harmful effects. The residual variability of responses to majority of the items between in- and outpatients was also different (residual variance non-invariance), implying substantial impact of health and social characteristics of in- and outpatients in assessing the beliefs about medicines. One should

also bear in mind that confirming configural, metric and scalar invariance is a prerequisite of a meaningful comparison of a scale scores between the tested groups [31]. Not meeting all these invariance criteria, no formal BMQ-PL score comparisons were performed.

The remaining psychometric properties of the scale were largely encouraging in the group of cardiovascular in- and outpatients and internal consistency was satisfactory. The mean score in each individual item was close to the middle value and the adequate dispersion was noted, suggesting the optimal use of the 5-point Likert scale and little skewness. Exploratory factor analysis together with association between the BMQ subscales, however, seem to provide confusing results. Although a 4-factor solution represented about 50% of data variability in factor analysis (as in the other reports [19, 43, 51]), and a clear separation of *Specific-Necessity* and *Specific-Concerns* subscales was found, the separation between *Specific* and *General* facets as well as *General-Overuse* and *General-Harm* subscales was substantially impaired. Again, the current results are not significantly different from those of previous studies. The original BMQ report [19] and many other BMQ language versions [43, 44, 46, 51] already indicated that *Specific* and *General* subscales are not fully separated, suggesting some overlap between *Specific-Concerns* with *General* items, which is coherent with present findings. The possible explanation of *General-Harm* and *Specific-Concerns* overlap is that the *General-Harm* beliefs about medicines may be formed from the personal experience of the patients, which is actually the representation of *Specific-Concerns* domain, as revealed by the interviews with the patients reported in this paper. The overlap between the *General-Overuse* and *General-Harm* was further illustrated by substantial association between the constructs and was discussed above. In fact, in the exploratory factor analysis, varimax rotation of the factors was used, which resulted in orthogonal factor presentation, falsely assuming no correlation between the constructs. Such a factor rotation method was also used by Mahler et al. [43]. It was employed to maximally differentiate the constructs, allowing compromising the quantitative interpretation of factor loading values.

The criterion-related validity of the BMQ-PL was fully confirmed in the group of cardiovascular outpatients, indicating the assumed association of beliefs about medicines with self-reported medication adherence. Stronger beliefs about the necessity of medicines was associated with better adherence, and stronger concerns and a more negative view of medicines with worse adherence. The present results support also the value of the well-defined concept of Necessity-Concerns Framework in predicting adherence [14]. Interestingly, such the associations were not found for the group of cardiovascular inpatients, apart from *Specific-Concerns* being negatively linked with adherence at borderline statistical significance. Some explanation for this phenomenon is provided by the residual analysis of regression modeling (Fig 3). For the group of inpatients, in contrast to outpatients, the distribution of residuals is far from normal, being highly left-skewed and compromising the estimate of regression parameters. This is because, in the group of inpatients, particularly those expressing lower *Necessity-minus-Concerns* beliefs, the results of self-reported adherence touched the maximum limit. Although the patients were informed that the results of the surveys (including the adherence questionnaire) will not be revealed to the healthcare practitioners, it cannot be excluded that the results are biased towards reporting more positive adherence [52] as a manifestation of observer-expectancy effect [53]. Another explanation is that a non-adherent patient may restart taking tablets (even discarding excessive ones [54]) in the pre-admission period (in case of planned admission) to appear to be following the regimen and report high adherence. Indeed, self-reported medication adherence measures are known to overestimate true adherence, and one factor predicting such overestimated adherence may be anxiety [52], which may increase in the pre-admission period [55, 56]. As a result, such false-positive pre-admission adherence may not be much influenced by individual beliefs about medicines. The association between beliefs about

medicines and drug adherence may be further complicated by the significant difference in the proportions of males and females between the inpatient and outpatient groups, as both the constructs may be related to sex [57, 58]. Consequently, the results of the association adjusted to covariates should be perceived as more accurate. An adherence measure that is more sensitive and less susceptible to bias [10] is needed to test for belief-adherence association in a well-balanced group of patients.

Contrary to the cardiovascular patients, the factorial structure of the construct was below borderline in the group of medical students as the fit indices were poor. Measurement invariance analysis showed that medical students may conceptualize the beliefs about medicines in a different way than patients without medical education or may attribute the other meaning to the items. Internal consistency estimates were heterogeneous across the subscales, being particularly low for the *General-Harm* subscale in medical students as expressed by Cronbach's alpha. Similar low values were replicated by some other reports suggesting the *General-Harm* subscale not being coherent for some groups of patients [19, 46], but here it may result from measurement non-invariance [42]. In response to the unsatisfactory psychometric properties of the original BMQ-PL structure for the *General* scale in the group of medical students, the modified BMQ-PL-General scale was proposed to be used in medical students and healthcare professionals (BMQ-PL-General-Med). The great advantage of the used methodology was that it employed a mixed method approach [59], blending qualitative semi-structured interviews with quantitative survey and psychometric analyses. This afforded an insight into the misconceptions that medically-educated people may hold about the BMQ. According to the authors' knowledge, such results were reported for the first time. Additionally, the external validation of the model confirmed the stability and applicability of the newly proposed BMQ-PL-General-Med scale.

The present study does have some limitations that warrant mention while interpreting the results. Firstly, the groups of patients were sampled with a convenience technique. This may restrict the generalizability of the results as it resulted in significant differences in the male-female composition of the inpatient and outpatient groups. Secondly, although the other research papers validating the BMQ imply stability of the construct representation between different groups of patients, no proof regarding the validity of the BMQ-PL was provided in other than cardiovascular and medical samples. Thus, one should be cautious, while applying BMQ-PL to the other groups of patients, ideally pre-testing the questionnaire each time. Thirdly, the selection of the statements to adapt the BMQ-PL for medically-educated people was limited to the set of 8 *General* statements included in the original BMQ-PL only, hence narrowing the spectrum of possible cognitive representations of the construct. Fourthly, the criterion-related validity was assessed with a self-reported measure of drug adherence only and, taking into account incoherence of the results, more objective outcome measures should be included. Finally, one should have in mind that the BMQ-PL was not formally validated as a decision-making tool to examine individual respondents in detail. Rather, the BMQ-PL should be used as a research tool to conclude about the phenomena observed at a population level. Consequently, using it to identify beliefs about treatment in a particular patient should be done with caution.

## Conclusion

The present report provides satisfactory proof of validity of the Polish version of Beliefs about Medicines Questionnaire to be used among cardiovascular patients. The obtained results confirm the value of the BMQ for medical and social research purpose. Medically-educated people may conceptualize the beliefs about medicines in general in slightly different way than laymen,

and the use of original BMQ for those medically educated is not recommended. Instead, a modified version of the BMQ-General, which appears suitable for medically educated people, is proposed. The BMQ-PL requires further validation procedures, however, the present report is conclusive and has a potential to push a research forward regarding beliefs about medicines.

## Supporting information

**S1 File. Understanding the BMQ-PL.**
(DOCX)

**S2 File. Measurement invariance between the tested groups of patients.**
(DOCX)

**S3 File. Further adaptation of the BMQ-General for the group of medically-educated people.**
(DOCX)

**S1 Table. Pearson's r and Spearman's rho coefficients for the associations.** Each row presents results of association between Variable 1 and Variable 2 in a given group of patients
(PDF)

**S1 Fig. Correlation between corresponding Pearson's r and Spearman's rho coefficients.**
The points represent corresponding coefficients given in S1 Table.
(TIF)

## Acknowledgments

We thank mgr. Joanna Bednarek, a specialist in Polish linguistics, for thorough review of the Polish version questionnaire, mgr. Edward Lowczowski, an English native speaker and an academic medical editor, for language assistance, prof. Adam Sagan for teaching the first author how to perform various types of confirmatory factor analyses, dr. Łukasz Mokros for his help in the Questionnaire translation and other colleagues for their valuable contribution. Above all, we thank all the respondents devoting their time to be interviewed or completing the questionnaire.

The copyright of the Polish version of the Beliefs about Medicines Questionnaire is allocated to prof. Robert Horne.

## Author Contributions

**Conceptualization:** Michał Seweryn Karbownik.

**Data curation:** Michał Seweryn Karbownik, Beata Jankowska-Polańska, Karol Maksymilian Górski.

**Formal analysis:** Michał Seweryn Karbownik.

**Funding acquisition:** Janusz Szemraj.

**Investigation:** Michał Seweryn Karbownik, Beata Jankowska-Polańska, Karol Maksymilian Górski.

**Methodology:** Michał Seweryn Karbownik.

**Project administration:** Michał Seweryn Karbownik.

**Resources:** Michał Seweryn Karbownik, Beata Jankowska-Polańska, Karol Maksymilian Górski.

**Supervision:** Robert Horne, Edward Kowalczyk, Janusz Szemraj.

**Validation:** Michał Seweryn Karbownik, Robert Horne.

**Visualization:** Michał Seweryn Karbownik.

**Writing – original draft:** Michał Seweryn Karbownik.

**Writing – review & editing:** Michał Seweryn Karbownik, Beata Jankowska-Polańska, Robert Horne, Karol Maksymilian Górski.

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
