## [Decision Letter · Decision Letter 0]

3 Feb 2020

PONE-D-19-34128

Adaptation and validation of the Polish version of the Beliefs about Medicines Questionnaire among cardiovascular patients and medical students

PLOS ONE

Dear Dr Karbownik,

Thank you for submitting your manuscript to PLOS ONE. After careful consideration, we feel that it has merit but does not fully meet PLOS ONE’s publication criteria as it currently stands. Therefore, we invite you to submit a revised version of the manuscript that addresses the points raised during the review process.

We would appreciate receiving your revised manuscript by Mar 19 2020 11:59PM. To enhance the reproducibility of your results, we recommend that if applicable you deposit your laboratory protocols in protocols.io, where a protocol can be assigned its own identifier (DOI) such that it can be cited independently in the future. For instructions see: http://journals.plos.org/plosone/s/submission-guidelines#loc-laboratory-protocols

We look forward to receiving your revised manuscript.

Kind regards,

Wen-Jun Tu

Academic Editor

PLOS ONE

Journal Requirements:

Reviewers' comments:

Reviewer's Responses to Questions

**Comments to the Author**

1. Is the manuscript technically sound, and do the data support the conclusions?

Reviewer #1: Partly

2. Has the statistical analysis been performed appropriately and rigorously? 

Reviewer #1: I Don't Know

3. Have the authors made all data underlying the findings in their manuscript fully available?

Reviewer #1: Yes

4. Is the manuscript presented in an intelligible fashion and written in standard English?

Reviewer #1: Yes

5. Review Comments to the Author

Reviewer #1: This is an overall well-planned, -conducted and -analyzed validation of a new version of an existing questionnaire. The need of an appropriate Polish instrument to measure beliefs about medicine is evident, the introduction is well-written. Moreover, it becomes clear that the authors invested a substantial amount of time and consideration into the development and testing of the questionnaire. The process of item forward- and back-translation and piloting appears appropriate. The discussion mentions potential limitations and is overall well-balanced. I have some comments / questions directed at guiding the reader to a better understanding of some analyses and their interpretations.

1. Although the authors employ and report a variety of analyses to demonstrate the reliability and validity of their instrument, at times, their use of criteria for how to judge outcomes as “good” or “acceptable” appears a bit imprecise. For example, to my knowledge, a scale has an acceptable consistency with a minimal Cronbach alpha of .7. However, the authors call internal consistency as “satisfactory (>0.6) for all the subscales in all the groups”. Similarly, they state that “The group of outpatients presented slightly worse model fit, not meeting conventional fit criteria; however, their fit may be regarded as acceptable.”(L.355). If they did not meet conventional fit criteria, what was the rationale behind regarding it as acceptable?

I would suggest that the authors either phrase these conclusions more cautiously, where given criteria are laid out but are not met on absolute terms (such as table 7, where they state criteria to judge the RMSEA), or clarify how they derive these interpretations (references, convention?).

2. With regards the model comparison of a 1 to 2 to 3 to 4 factor model, it seems that the authors aimed to replicate the 4-factor model, yet it is not always clear which model criteria they employed for this decision. In some regards, the 3-factor model was favored ("Moreover, parsimonious fit index and information criteria favored the simpler, 3-factor model, over the original 4-factor model in both in- and outpatients.", L. 352), however the authors accept confirmation of the 4-factor structure. Although they elaborate on this in the discussion, and it replicates findings from other versions of the BMQ, I would advise to report more clearly (in the results section), which model criteria lead to the conclusion that "little evidence against" the 4-factor model makes it "highly advisable" (L.611).

3. After studying differences between subgroups, the authors employ a 2-factor-model in the subgroups, and a 4-factor-model in the pooled sample: “The analysis was performed separately for Specific and General scales in each group of respondents (Table 6) and together for all the subscales in the pooled sample (Table 7)”. The motivation to split the analysis for Specific and General scales is not clear to me. Moreover, why is the approach not held constant for the pooled sample (specific and general separately)? Or alternatively, why apply a 4-factor model to the pooled sample, if shown before, that groups differ in how they respond to the items? It would be beneficial if the authors could clarify these approaches to guide the reader to a better understanding of their analyses.

4. It is mentioned that “Although the Likert scale data used in the questionnaires should be perceived as ordinal variables, the associations between the examined constructs were assessed with Pearson correlation coefficient (r) and were further explored with general linear modeling (GLM), i.e. parametric tests. This was to allow for multivariate modeling. However, corresponding non-parametric tests were performed where possible, and all yielded very similar results to the parametric ones.”

This is somehow contradictory – if the Likert scale is considered ordinal, I would expect the use of Spearman’s rho (that is mentioned to result in “very similar results”, which are however not reported), or more information on why this approach is feasible.

Minor:

1. L. 352: ‘parsimonious fit index and information criteria‘: Could you please specify the criteria you are referring to, e.g. BIC/AIC Table 3?

2. The proportion of male/female participants differs between groups, especially between in- and outpatients, where the majority of outpatients were female. I would suggest to elaborate on this further, or at least add this to the discussion as potential limitation, e.g. with regards to the generalizability of the findings.

6. PLOS authors have the option to publish the peer review history of their article (what does this mean?). If published, this will include your full peer review and any attached files.

Reviewer #1: No

---

## [Author Response · Author response to Decision Letter 0]

13 Feb 2020

Responses to Editor and Reviewers

Below we list the responses to the Editor’s and Reviewers’ comments.

Editor’s comments

We have made every effort to ensure that our manuscript meets PLOS ONE’s style requirements, according to the provided template. We hope that we have included all necessary corrections.

The DOI number of the repository has been already provided (line 269 in the revised manuscript), however, it has not been activated yet. We will activate it once our manuscript is accepted for publication.

Originally we wrote: “Some explanation for this phenomenon is provided by the residual analysis of regression modeling (data not shown).”

By “data not shown” we meant “the results of the analysis were not explicitly reported”. In fact, data underling this analysis will be publicly available in a data repository (should our manuscript will be accepted for publication) as stated in lines 267-269 (according to the revised manuscript), but the result of residual analysis were not explicitly reported in the manuscript. The only presentation of residual analysis is included in Figure 3.

Consequently, the phrase “data not shown” has been replaced with “(Figure 3)”.

Reviewer's comments

Reviewer #1: This is an overall well-planned, -conducted and -analyzed validation of a new version of an existing questionnaire. The need of an appropriate Polish instrument to measure beliefs about medicine is evident, the introduction is well-written. Moreover, it becomes clear that the authors invested a substantial amount of time and consideration into the development and testing of the questionnaire. The process of item forward- and back-translation and piloting appears appropriate. The discussion mentions potential limitations and is overall well-balanced. I have some comments / questions directed at guiding the reader to a better understanding of some analyses and their interpretations.

We thank Reviewer #1 for the careful reading of our manuscript and for his/her constructive comments.

1. Although the authors employ and report a variety of analyses to demonstrate the reliability and validity of their instrument, at times, their use of criteria for how to judge outcomes as “good” or “acceptable” appears a bit imprecise. For example, to my knowledge, a scale has an acceptable consistency with a minimal Cronbach alpha of .7. However, the authors call internal consistency as “satisfactory (>0.6) for all the subscales in all the groups”. Similarly, they state that “The group of outpatients presented slightly worse model fit, not meeting conventional fit criteria; however, their fit may be regarded as acceptable.”(L.355). If they did not meet conventional fit criteria, what was the rationale behind regarding it as acceptable?

I would suggest that the authors either phrase these conclusions more cautiously, where given criteria are laid out but are not met on absolute terms (such as table 7, where they state criteria to judge the RMSEA), or clarify how they derive these interpretations (references, convention?).

In the revised version of our manuscript, Cronbach’s alpha ≥0.7 is now labelled as “acceptable”, whereas ≥0.6 as “questionable” according to Crutzen and Peters (2017); the Results section has been modified accordingly (lines 437-440 in the revised manuscript). Although this convention to label alpha values has been widely accepted, this is not based on empirical research or logical reasoning (Hoekstra 2019). Moreover, as expressed by Schmitt (1996): “There is no sacred level of acceptable or unacceptable level of alpha. In some cases, measures with (by conventional standards) low levels of alpha may still be quite useful.” Consequently, as advised by Taber (2018), the Discussion section has been supplemented with comments on the interpretability of Cronbach’s alpha (lines 613-624 in the revised manuscript). We also appreciated McDonald’s omega in estimation of internal consistency in the above mention comments. As McDonald’s omega presents generally more favorable results than Cronbach’s alpha in our research, the term “satisfactory” was kept in the Abstract, Results and Discussion sections (lines 35, 558 and 650) to describe a more “general” (i.e. composed of Cronbach’s alpha and McDonald’s omega) estimate of internal consistency.

To be more precise and less speculative in describing the results of confirmatory factor analysis (CFA) model fitting, it was stated whether a model meets conventional fit criteria or not (see lines 385-386 in the revised manuscript). Some expressions have been adjusted to provide a more precise description of whether the conventional model fit criteria were met (lines 408-409 and 626-629 in the revised manuscript).

2. With regards the model comparison of a 1 to 2 to 3 to 4 factor model, it seems that the authors aimed to replicate the 4-factor model, yet it is not always clear which model criteria they employed for this decision. In some regards, the 3-factor model was favored ("Moreover, parsimonious fit index and information criteria favored the simpler, 3-factor model, over the original 4-factor model in both in- and outpatients.", L. 352), however the authors accept confirmation of the 4-factor structure. Although they elaborate on this in the discussion, and it replicates findings from other versions of the BMQ, I would advise to report more clearly (in the results section), which model criteria lead to the conclusion that "little evidence against" the 4-factor model makes it "highly advisable" (L.611).

Our decision to support the 4-factor model of BMQ-PL was based on 1) experimental data and 2) existing literature. 

1) Parsimony-corrected Comparative Fit Index as well as Akaike’s and Bayesian Information Criteria favored the simpler, 3-factor model in the group of in- and outpatients. However, the differences were so minimal that appeared insignificant and could potentially disappear with follow-up study.

2) Majority of the cross-cultural BMQ validation literature supports the use of the 4-factor model, including such languages as German (Mahler et al. 2010) or Czech (Matoulkova et al. 2013), which are spoken in countries bordering Poland and which are culturally similar to Poland.

As no solid evidence was found against two-dimensional General beliefs about medicines represented by Polish people, the original 4-factor model of the BMQ-PL was preserved.

Suitable information about this was added to the Results section (see lines 351-358 in the revised manuscript).

3. After studying differences between subgroups, the authors employ a 2-factor-model in the subgroups, and a 4-factor-model in the pooled sample: “The analysis was performed separately for Specific and General scales in each group of respondents (Table 6) and together for all the subscales in the pooled sample (Table 7)”. The motivation to split the analysis for Specific and General scales is not clear to me. Moreover, why is the approach not held constant for the pooled sample (specific and general separately)? Or alternatively, why apply a 4-factor model to the pooled sample, if shown before, that groups differ in how they respond to the items? It would be beneficial if the authors could clarify these approaches to guide the reader to a better understanding of their analyses.

The BMQ validation literature is inconsistent with the way of presenting the results of exploratory factor analysis (EFA). In order to give some examples:

• Horne et al. (1999), while developing the original tool, presented separate EFAs for Specific and General scales for a pool of selected patients as well as a single EFA with all the four subscales for a pooled sample of all the patients.

• Mahler et al. (2010), while validating the German BMQ version, used only a single EFA with all the four subscales for all the patients.

• Komninos et al. (2012), while validating the Greek BMQ version, applied only one type of EFA: separate for Specific and General scales for all the patients.

• Matoulkova et al. (2013), while validating the Czech BMQ version, replicated the approach of Horne et al. (1999) to use both types of EFA: separate for Specific and General scales for all the patients as well as a single EFA with all the four subscales for all the patients.

• Fall et al. (2014), while validating the French BMQ version, used no EFA at all.

• Gatt et al. (2017), while validating the Maltese BMQ version, applied only one type of EFA: a single EFA with all the four subscales for all the patients representing different illnesses.

In the original version of our manuscript we adapted the “richest” version of EFA: 1) separately for Specific and General scales for each groups of patients as well as 2) a single EFA with all the four subscales for a polled sample of patients; see: Horne et al. (1999) and Matoulkova et al. (2013). However, pooling all the patients lead us to a dead end, as multi-group CFA (MGCFA) indicated that the groups of patients may conceptualize BMQ-PL in different ways, and should not be combined as a result. On the other hand, it could be too confusing to report three independent EFAs with all the four subscales and the results would lack real value, as the associations between subscales of Specific and General scales have already been presented in Table 8 (numbering according to the original version of the manuscript).

As our results are already “rich” in content we decided to retain Table 6 (separate EFA for Specific and General scales for separate groups of patients), while discarding Table 7 (a single EFA with all the four subscales for a polled sample of patients) to enhance the clarity of the revised manuscript.

4. It is mentioned that “Although the Likert scale data used in the questionnaires should be perceived as ordinal variables, the associations between the examined constructs were assessed with Pearson correlation coefficient (r) and were further explored with general linear modeling (GLM), i.e. parametric tests. This was to allow for multivariate modeling. However, corresponding non-parametric tests were performed where possible, and all yielded very similar results to the parametric ones.”

This is somehow contradictory – if the Likert scale is considered ordinal, I would expect the use of Spearman’s rho (that is mentioned to result in “very similar results”, which are however not reported), or more information on why this approach is feasible.

Parametric tests are more welcome in multidimensional analyses than non-parametric ones as they give more analytical potential (such as adjusting to covariates or testing interaction effects). There has been a long-lasting heated debate on whether parametric tests can be used to analyze Likert scale data, which is in fact ordinal. Thorough review of available literature (Carifio and Perla 2007, Mircioiu and Atkinson 2017, Glass et al. 1972) suggests that the parametric tests could be applied for majority of Likert scale data analyses. Nevertheless, two auxiliary analyses were run to test whether our parametric results are robust: 1) tests of normal distribution of residuals and 2) non-parametric equivalent test.

1) Tests of normal distribution of residuals. In majority of correlation analyses the assumption of normal distribution of residuals was not significantly violated. The only severe violation was noted for the association between self-reported drug adherence and necessity, concern and necessity-minus-concern beliefs about medicines in the group of inpatients. This issue has been already discussed in the original version of the manuscript (see lines 679-688 in the revised manuscript).

2) Non-parametric equivalent analyses. As already stated in “Data analysis” section of our manuscript, we run non-parametric equivalent tests, wherever possible, to check for similarities with corresponding parametric ones. In response to the Reviewer’s comment, the revised version of the manuscript also reports the results of non-parametric tests together with parametric ones in table and graph form (S1 Table and S1 Figure in revised manuscript). The corresponding non-parametric and parametric association coefficients were very similar in all cases but two; this was acknowledged in the revised version of the manuscript (see Table 7 and 8 and lines 548-549 in the revised manuscript). 

Minor:

1. L. 352: ‘parsimonious fit index and information criteria‘: Could you please specify the criteria you are referring to, e.g. BIC/AIC Table 3?

The indices were specified (see lines 349-350 in the revised manuscript).

2. The proportion of male/female participants differs between groups, especially between in- and outpatients, where the majority of outpatients were female. I would suggest to elaborate on this further, or at least add this to the discussion as potential limitation, e.g. with regards to the generalizability of the findings.

The results of the χ2 test regarding the difference in the proportion of males to females across the groups of participants was added to the footnote of Table 2. This issue was briefly discussed in the Discussion section (see lines 694-697 and 700 in the revised manuscript) and summarized in the Limitation subsection of Discussion (see lines 720-722 in the revised manuscript) as advised by the Reviewer.

References

Carifio J, Perla RJ. Ten common misunderstandings, misconceptions, persistent myths and urban legends about Likert scales and Likert response formats and their antidotes. J Social Sci. 2007;3:106-116.

Crutzen R, Peters GY. Scale quality: alpha is an inadequate estimate and factor-analytic evidence is needed first of all. Health Psychol Rev. 2017;11:242-247.

Fall E, Gauchet A, Izaute M, Horne R, Chakroun N. Validation of the French version of the Beliefs about Medicines Questionnaire (BMQ) among diabetes and HIV patients. Eur Rev Appl Psychol. 2014;64:335-343.

Gatt I, West LM, Calleja N, Briffa C, Cordina M. Psychometric properties of the Belief about Medicines Questionnaire (BMQ) in the Maltese language. Pharm Pract (Granada). 2017;15:886.

Glass GV, Peckham PD, Sanders JR. Consequences of failure to meet assumptions underlying the analysis of variance and covariance. Rev Educ Res. 1972;42:237-288.

Hoekstra R, Vugteveen J, Warrens MJ, Kruyen PM. An empirical analysis of alleged misunderstandings of coefficient alpha. Int J Soc Res Methodol. 2019;22:351-364.

Horne R, Weinman J, Hankins M. The Beliefs about Medicines Questionnaire: the development and evaluation of a new method for assessing the cognitive representation of medication. Psychol Health. 1999;14:1-24.

Komninos ID, Micheli K, Roumeliotaki T, Horne R. Adaptation and validation of the Beliefs about Medicine Questionnaire (BMQ) in primary care patients in Greece. Eur J Pers Cent Healthc. 2012;1:224-231.

Mahler C, Hermann K, Horne R, Jank S, Haefeli WE, Szecsenyi J. Patients' beliefs about medicines in a primary care setting in Germany. J Eval Clin Pract. 2012;18:409-13.

Matoulkova P, Krulichova IS, Macek K, Vlcek J, Bastecka D, Prixova M, et al. Chronically ill Czech patients' beliefs about medicines: the psychometric properties and factor structure of the BMQ-CZ. Ther Innov Regul Sci. 2013;47:341-348.

Mircioiu C, Atkinson J. A comparison of parametric and non-parametric methods applied to a Likert scale. Pharmacy (Basel). 2017;5: pii: E26.

Schmitt N. Uses and abuses of coefficient alpha. Psychol Assess. 1996;8:350-353.

Taber KS. The use of Cronbach’s alpha when developing and reporting research instruments in science education. Res Sci Educ. 2018;48:1273-1296.

---

## [Editor Report · Decision Letter 1]

24 Feb 2020

Adaptation and validation of the Polish version of the Beliefs about Medicines Questionnaire among cardiovascular patients and medical students

PONE-D-19-34128R1

Dear Dr. Karbownik,

We are pleased to inform you that your manuscript has been judged scientifically suitable for publication and will be formally accepted for publication once it complies with all outstanding technical requirements.

With kind regards,

Wen-Jun Tu

Academic Editor

PLOS ONE
---

## [Editor Report · Acceptance letter]

3 Mar 2020

PONE-D-19-34128R1 

Adaptation and validation of the Polish version of the Beliefs about Medicines Questionnaire among cardiovascular patients and medical students 

Dear Dr. Karbownik:

I am pleased to inform you that your manuscript has been deemed suitable for publication in PLOS ONE. Congratulations! Your manuscript is now with our production department. 

With kind regards,

on behalf of

Dr. Wen-Jun Tu 

Academic Editor

PLOS ONE